# Hyo-Mental Angle and Distance: An Important Adjunct in Airway Assessment of Adult Mucopolysaccharidosis

**DOI:** 10.3390/jcm10214924

**Published:** 2021-10-25

**Authors:** Chaitanya Gadepalli, Karolina M. Stepien, Govind Tol

**Affiliations:** 1Ear Nose and Throat Department, Salford Royal NHS Foundation Trust, Manchester M6 8HD, UK; 2Adult Inherited Metabolic Department, Salford Royal NHS Foundation Trust, Manchester M6 8HD, UK; karolina.stepien@srft.nhs.uk; 3Anaesthetics Department, Salford Royal NHS Foundation Trust, Manchester M6 8HD, UK; govind.tol@srft.nhs.uk

**Keywords:** mucopolysaccharidosis, airway management, radiology, hyoid bone, chin, intubation, intratracheal

## Abstract

Background: Mucopolysaccharidosis (MPS) is a rare congenital lysosomal storage disorder with complex airways. High anterior larynx is assessed by thyromental distance (TMD) nasendoscopy. A simpler method to assess this hyoid bone is described. The distance between the central-hyoid and symphysis of the mandible (hyo-mental distance; HMD) and inclination of this line to the horizontal axis (hyo-mental angle; HMA) in neutrally positioned patients is investigated. Methods: HMA, HMD in MPS, and non-MPS were compared, and their correlation with height and weight were assessed. Results: 50 adult MPS patients (M = 32, F = 18, age range = 19–66 years; mean BMI = 26.8 kg/m^2^) of MPS I, II, III, IV, and VI were compared with 50 non-MPS (M = 25, F = 25; age range = 22–84 years; mean BMI = 26.5 kg/m^2^). Mean HMA in MPS was 25.72° (−10 to +50) versus 2.42° (−35 to +28) in non-MPS. Mean HMD was 46.5 (25.7–66) millimeters in MPS versus 41.8 (27–60.3) in non-MPS. HMA versus height and weight showed a moderate correlation (r = −0.4, *p* < 0.05) in MPS and no significant correlation (r < 0.4, *p* > 0.05) in non-MPS. HMD versus height and weight showed no correlation (r < 0.4, *p* > 0.05) in both groups. Conclusions: HMA seems more acute in MPS despite nearly the same HMD as non-MPS, signifying a high larynx, which may be missed by TMD.

## 1. Introduction

Mucopolysaccharidosis are a group of inherited congenital multisystem diseases due to a deficiency in enzymes required for the breakdown of complex mucopolysaccharides. Mucopolysaccharidoses (MPSs) are rare, inherited, lysosomal storage diseases with a combined incidence of 1 in 22,000 [1]. The disease is characterized by the accumulation of glycosaminoglycans (GAGs) in almost all parts of the body. There are seven types of MPS depending on the type of enzyme deficiency (Table 1). The manifestations of this disease are multisystemic, resulting in shortened longevity [2]. Along with other systems, airways are commonly involved. Knowledge about the airway abnormalities is important as these patients can pose airway problems. With advances in treatment modalities, such as hematopoietic stem cell transplantation (HSCT) [3] and enzyme replacement therapy (ERT) [4], the longevity of these patients has increased posing newer problems. Most of these patients will need general anesthetic for surgery at some point in their lifetime due to multi-system involvement. In our experience, we noted that most adult MPS patients have a large and bulky tongue, large lower jaw, and short neck [5]. A high or anterior larynx poses difficulty in access to the airway. A difficult laryngoscopy is defined as an inability to visualize any part of the vocal cords on conventional laryngoscopy [6]. Various bedside measures have been described to assess a high or anterior larynx. The method commonly used to assess a high or anterior larynx are measurement of the thyromental distance TMD [7], nasendoscopy, or cross-sectional imaging. TMD is the distance between the chin, also called the mentum, and prominence of the thyroid cartilage with the neck extended. TMD [7] is about 6.5 cm measured with an intubation gauze or three finger breadths. Direct visualization of the oropharynx and larynx via a fiber optic scope, also called nasendoscopy, can also estimate a high larynx. Figure 1 shows a nasendoscopy picture of a patient with a normal larynx. Figure 2 shows a nasendoscopy picture in an MPS I patient and Figure 3 shows a nasendoscopy picture in an MPS II patient. Cross-sectional imaging either with a computer tomography scan (CT) or magnetic resonance imaging (MRI) provides a detailed evaluation of the head and neck in a neutral position. Many times, these scans have been performed in MPS patients for their co-morbidities. We propose a simpler and possibly more accurate method of assessing a high or anterior larynx using existing cross-section imaging of the neck. This information is very helpful in planning any airway intervention or general anesthetic. We use the body of the hyoid bone as a stable landmark: the distance from the center of the body of hyoid to the symphysis mentum of the mandible is the hyomental distance (HMD). The inclination or declination of HMD to the horizontal axis is the hyomental angle (HMA). The HMD quantifies the anterior larynx and HMA identifies the high larynx. Figure 4 shows a sagittal section diagram showing the landmarks and Figure 5 shows HMA and HMD on a CT scan in a non-MPS and MPS patient. Figure 6 is a three-dimensional reconstruction of the upper body skeleton, showing the level of the hyoid bone in a non-MPS and MPS patient in relation to the mandible. The primary aim of this study was to identify a different and possibly a more accurate method to assess a high or anterior larynx and discuss its usefulness in airway management in adult MPS. The secondary aim was to assess the relationship of the body habitus to a high and anterior larynx.

## 2. Materials and Methods

Retrospective analysis of case notes and radiological investigations was performed as part of routine care of 50 adult MPS patients and comparison with 50 healthy adults of similar age, gender, and body mass index (BMI). Ethical approval was obtained from the local research and innovation department, Salford Royal NHS Foundation Trust, Northern Care Alliance, United Kingdom, reference: S20HIP40. HMD and HMA were calculated for each group calculated in the picture archiving communications system (PACS) using the ruler tool. A smaller HMD was considered to be an anterior larynx and acute inclination of the HMA to the horizontal axis was considered to represent a high larynx. Both HMD and HMA reflect the difficulty in accessing an airway. The impact of body habitus on a high or anterior larynx was investigated by calculating the Pearsons correlation between HMD, HMD, and weight and height.

## 3. Results

Radiological cross-sectional images of 50 MPS and 50 non-MPS patients were included in the study. The MPS group included patients with types I, II, III, IV, and VI. Table 2 depicts the demographics in both groups. It may be noted that even though age does not match between both groups, BMI is comparable. The non-MPS group of patients included a range of patients of various ENT (ear, nose, and throat) pathologies who had imaging studies. All the patients in the non-MPS group had no pathology in the oral cavity, neck, oropharynx, supraglottis, and hypopharynx. The non-MPS group had a normal supra-glottic airway. This enabled us to investigate the abnormal hyomental region in MPS patients. Table 3 depicts the pathology subtypes in both groups. The HMA and HMD were calculated and compared between the two groups. 

Table 4 depicts HMA and HMD in both MPS and non-MPS groups. It may be noted that the HMD is slightly less in the MPS depicting slightly anterior larynx in the MPS group. The HMA is more acute in the MPS group compared to the non-MPS group, depicting that larynx is higher in the MPS groups. The MPS group have a shorter stature and lower body mass, this is a recognized feature of MPS due to multisystemic involvement of the disease.

It can be assumed that a bulky upper airway may be attributable to BMI, thereby affecting HMA or HMD. To test this hypothesis, the Pearson correlation between the BMI versus HMA and HMD was calculated and Table 5 depicts the results. HMD shows no correlation with height, weight, and BMI in the non-MPS group but reveals a statistically significant correlation with height and weight in MPS (*p* = 0.05; *p* = 0.009). It must be noted that the rho value in the MPS group is only 0.3 at best. The Pearson correlation between HMA versus height and weight showed a moderate negative correlation in the MPS group and no correlation in the non-MPS group. Thus, HMA shows a better correlation with height and weight in the MPS group, compared to HMD. The significant results are highlighted in the table. 

## 4. Discussion

### 4.1. Difficult Airway

Airway complications are a common feature of MPS I, II, IV, and VI and considerably contribute to morbidity and premature mortality [11,12]. Airway assessment is ideally performed holistically, taking into account all the factors in the upper and lower airways with various methods, including medical history, clinical examination, radiological evaluation, and endoscopy [13]. A high and anterior larynx is one of the important aspects in the upper airway, which can lead to difficult intubation due to poor access, also called difficult laryngoscopy. Failure to recognize airway problems pre-operatively or during the planning of airway intervention can lead to unfavorable outcomes. MPS is a rare disease, and awareness amongst health professionals regarding adult MPS patients is poor. Once a patient is paralyzed and anaesthetized, the tongue falls backwards, and the oropharynx collapses inwards. In this situation, a high or an anterior larynx makes access to the larynx difficult if not impossible. In a patient who is paralyzed and anaesthetized, this can lead to situation of “cannot intubate- cannot ventilate”. The Difficult Airway Society (DAS; UK) has produced guidelines on this difficult situation [14]. This difficult situation can be prevented by recognition of the problem of difficult access by existing cross-section images. Metanalysis of 35 studies representing 50,760 patients revealed the incidence of difficult intubation is about 5.8 % in normal patients, 3.1% in obstetric patients, and 14.8 in obese patients [15]. This may be higher in MPS due to deposition of GAGs in the soft tissues and musculoskeletal system, leading to bulky upper airways and bony abnormalities. A combination of abnormalities in the soft tissue, cervical spine, and skull leads to a high larynx and anterior larynx, resulting in difficulty accessing the airway. In our study, an attempt was made to match the MPS and non-MPS groups. It is not possible to obtain an exact match as age-related changes are faster in the MPS group with a shortened life span. Most of the MPS patients have short stature; however, it can be noted that both MPS and non-MPS groups have nearly similar BMI. We must also understand that MPS patients have a short stature and have truncal obesity [16], and non-MPS patients in our group are taller. Hence, BMI may be a misleading airway health measure in MPS patients. The PACS has in-built tools to measure the distances and angles in the cross-sectional images. As this measurement tool is computerized, we can assume that inter or intra-rater variability is reduced. Bias may be observed if the landmarks are not correctly identified by the clinician. In situations where the clinician does not have access to PACS, an angle measure and a ruler can be used to obtain HMD or HMA on existing cross-sectional imaging. In our study, we note that HMD was slightly less in the MPS group, indicating that the larynx is mildly anterior. The overall difference, however, in HMD in the MPS and non-MPS groups is minimal. On the other hand, HMA was more acute in the MPS group, indicating that the larynx is higher in the MPS group. It is interesting to note that HMA can vary in the MPS group despite nearly the same HMD as non-MPS. 

Moreover, it was observed that some MPS patients did not have acute HMA and nearly the same HMD as non-MPS. The reasons for this could be multifactorial. Firstly, 16 results from MPS type I had milder upper airway abnormalities; secondly, the severity of MPS is dependent on mutations, the length, or their therapies, such as enzyme replacement therapy (ERT) or hematopoietic stem cell transplantation (HSCT). ERT in MPS I Hurler–Scheie (HS) and Scheie, II, IVA, and VI and HSCT in MPS I Hurler (H) have demonstrated organ-specific and systemic metabolic correction [17,18,19,20]; hence, the severity of the disease is variable. Advances in treatment strategies have improved life expectancy, and the average age of our cohort was 31.7 years. So, our study may be representative of varied MPS phenotypes and younger adult MPS patients. 

### 4.2. Clinical Measures of Difficult Airway

TMD is a commonly used tool in airway assessment. The sensitivity of TMD is about 25% (95% confidence interval: 23–28) and specificity is 90.2% (95% confidence interval: 90–91) [21]. In MPS patients, the lower jaw may be disproportionately large, resulting in a normal thyromental distance despite a high and anterior larynx. TMD may also be of limited use in patients with facial and skeletal dysmorphism, and bulky soft tissues of the neck and sub mental region as commonly noted in adult MPS. The sternomental distance (SMD) [22] is calculated by measuring the distance between the mentum and the manubrium sternum with the mouth closed. It may indicate the degree of neck extension, which is important in access to the airway. The authors [22] conclude that a sternomental distance of 13.5 cm or less was 66.7% sensitive and 71.1% specific, and the positive and negative predictive values were 7.6% and 98.4%, for difficult laryngoscopy [22]. In their study, there was no association between sternomental distance and age, weight, height, or BMI. Sternomental distance may indirectly reflect a high larynx, and further research can be done to invesitgate this association. The TMD and SMD may be normal in MPS patients due to a large lower jaw, giving a false sense of security of a normal airway. The distance between the hyoid and thyroid prominence may not be as variable, so HMD and TMD may represent the same measure. HMD may be more accurate as it does not take into account the sub cutaneous soft tissue of the neck. In our study, we observed that the HMD is nearly the same in the MPS and non-MPS groups but the HMA was more acute. This may indicate that HMA is a more accurate measure of a high or anterior larynx than HMD or TMD. The results of this study showed that there is no correlation between the HMA and HMD in both the MPS and non-MPS groups, suggesting that HMA may be a completely independent entity, not related to the distance between the mentum to the laryngeal framework. The HMA correlates negatively with height and weight but has no correlation with BMI. This is because correlation is a linear measure and BMI is weight divided by height squared. Hence, observing the weight and height independently may be more useful than BMI in airway assessments. This may be more relevant to the MPS population as they are known to have short stature and central obesity [23,24]. The other commonly used bedside airway assessments methods are neck movements, neck circumference, Wilson’s score [25], mallampati [26], and modified mallampati grade [27]. Mallampati and modified mallampati grade assess the size of the tongue in relation to the opened oral cavity. Dalewski [28] et al., in a study of 129 adults, suggested the combination of Mallampati grade, CT scan upper airway volume, and Berlin score to calculate snoring and breathlessness. The authors noted a positive correlation between high modified mallampati grade, BMI, and reduced oxygen saturations and upper airway volume. The pre-operative assessment aims to assess the difficult airway and plan a difficult situation. Based on laryngoscopy views, Cormack [29] graded the airway into three grades: grade 1 being full view of the glottis, grade 2—partial view of the glottis, grade 3—only epiglottis is visible, and grade 4—neither epiglottis nor glottis are visible. Modifications [25,30] of grade 2 resulted in 2a being part of the glottis visible and 2b being arytenoids or posterior cords only just visible. Cook [31] suggested the grading system as “E” as easy view—grade 1 and 2a “R” restricted view—grade 2b and 3a and “D” difficult view—grade 3b and 4, where the grade being epiglottis can be elevated with a gum elastic bougie and 3b being epiglottis cannot be elevated by a gum elastic bougie. Knowledge of a grade R or D prior to intubation in the pre-assessment clinics is very useful; HMD and HMA will provide this information. Radiology plays an important role in airway assessment for an anesthetist [32]. Although lateral radiographs, chest X-rays, and ultrasonography [33,34] are useful, in our experience, we found that the use of MRI scans is helpful in upper airways and CT scans in lower airways. The images from CT and MRI scans can be used to perform three-dimensional reconstruction of the airways in MPS [35] and perform virtual endoscopy. We also found nasendoscopy to be very useful in adult MPS [36]. Imaging of the airway is not routinely performed for unsuspected airway problems. Any additional investigations should be carefully considered, keeping patient comfort in mind. Although HMD has been reported [37] as a predictor of difficult airway in patients with cervical spondylosis, HMA, to the best of our knowledge, has not been reported so far. We feel that this easily available tool is an adjunct to airway assessment and can be adopted in adult MPS and any other difficult airway situation. In our personal experience of adult airway assessment, we feel HMA close to zero or less than zero indicates a larynx that is not high. HMA could be considered as another important adjunct in upper airway assessment; however, holistic airway assessment should include both upper and lower airways. 

### 4.3. Limitations of the Study

#### 4.3.1. Head and Neck Position

Most of our MPS patients had cervical spine issues so we chose to take radiological images at the neutral position of the head, keeping patient comfort as the priority. To keep the upper airway open, the natural instinct of any patient is to adopt a sniffing posture. In all our MPS patients, some form of airway and cervical spine abnormality was noted, which may have prompted patients to adopt a comfortable posture. These factors could have skewed some of our measurements. It may be argued that HMA, which is acute in the neutral comfortable position in MPS compared to non-MPS, may be more acute in a standard position of the head and neck. Future studies could include standardization of head positions to obtain radiological images to obtain accurate measures of the HMA and HMD to test this hypothesis. Extension of the neck will improve laryngoscopic views. Future studies could also incorporate measurement of HMA and HMD in maximum extension and comparison with HMA and HMD in the neutral position to assess the degree of improvement of laryngoscopy views by neck extension.

#### 4.3.2. Thyro-Hyoid Distance

We made the assumption that the distance between the hyoid and thyroid is small enough to assume that HMD and TMD reflect the same measure of a high or anterior larynx. Future studies could also assess the thyrohyoid distance in flexion and extension of the neck to test this hypothesis. 

#### 4.3.3. Numbers

Our cohort examined only 50 adult MPS patients of various types and varying severity, which was compared with 50 adults with no upper airway issues. Considering the rarity of the disease, this may appear a significant number; however, a larger study group could have produced more significant results. Future studies may incorporate larger number of patients in both pediatric and adult MPS by a multi-center collaboration. 

### 4.4. Wider Implications

HMD and HMA application can be extended to wider use of difficult airway assessment in any patient due to its simplicity in use. This may play a special role in those with cervical spine or any craniofacial anomalies. 

## 5. Conclusions

HMA and HMD are useful measurements that can be obtained from existing cross-section imaging, providing important information about an anterior or high larynx. This is very helpful in pre-planning during airway assessment as part of the pre-operative work-up. HMA may be a better indicator than HMD in MPS patients. The use of HMA and HMD can be extrapolated to airway assessments in other patients with or without head and neck dysmorphism. This simple airway assessment tool is a useful adjunct in the management of complex airways, such as adult MPS. Further investigation into the sensitivity and specificity of HMA and HMA with a standardized head position and its correlation with difficult intubation will be useful.

## Figures and Tables

**Figure 1 jcm-10-04924-f001:**
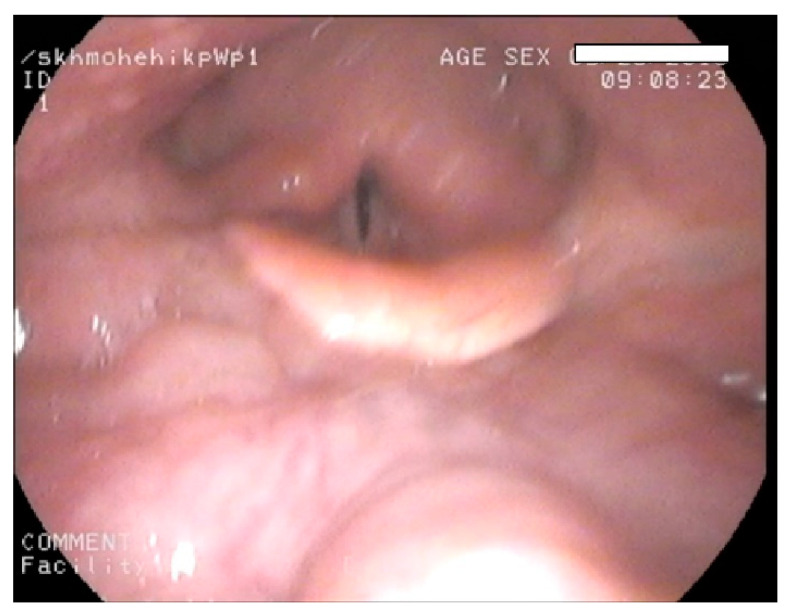
Nasendoscopy pictures showing a normal larynx: the vallecula, epiglottis, glottic inlet, and posterior larynx can be clearly seen.

**Figure 2 jcm-10-04924-f002:**
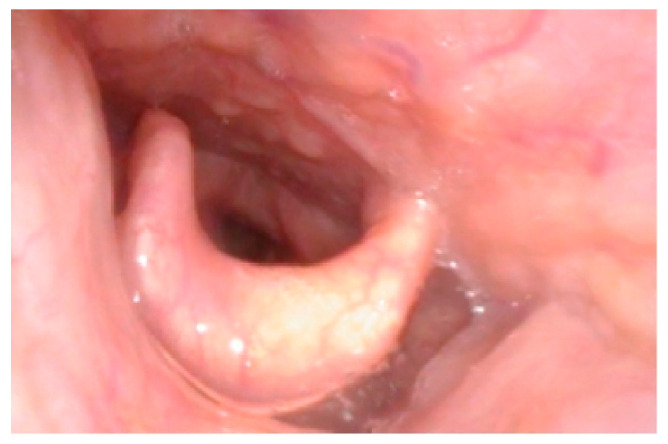
Nasendoscopy picture in MPS I showing a high larynx, large epiglottis touching the soft palate, vallecula cannot be seen, and only the posterior glottis is visible.

**Figure 3 jcm-10-04924-f003:**
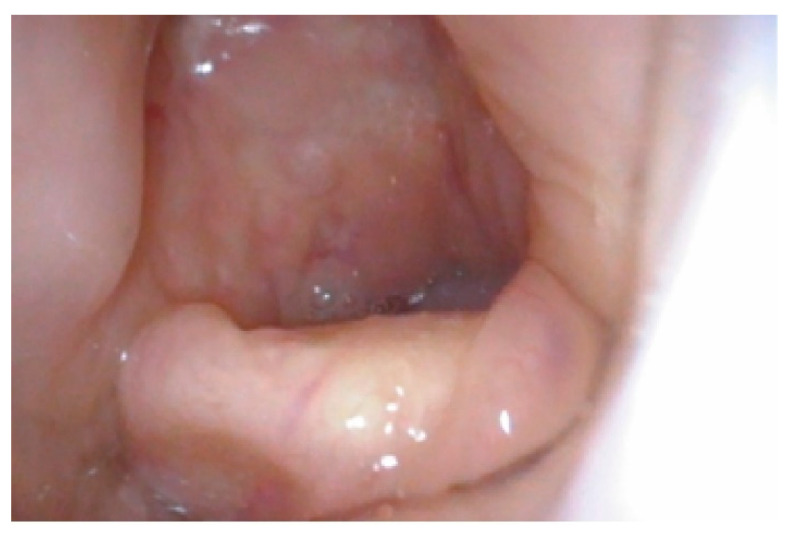
Nasendoscopy picture of MPS II showing a high and anterior larynx, epiglottis is touching the soft palate, vallecula is not seen, and only posterior pharyngeal wall is seen.

**Figure 4 jcm-10-04924-f004:**
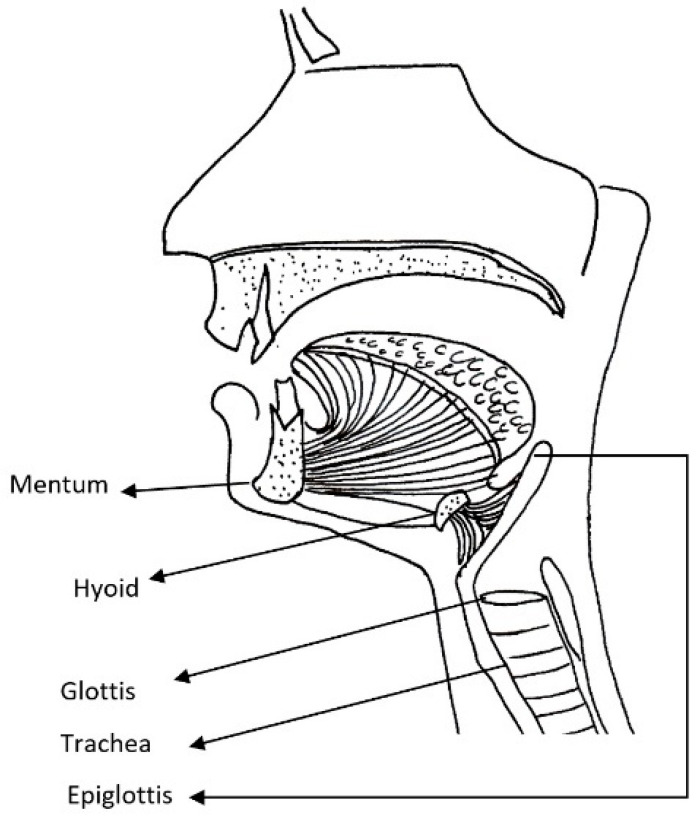
Line diagram sagittal section of the head and neck showing the upper airway and landmarks.

**Figure 5 jcm-10-04924-f005:**
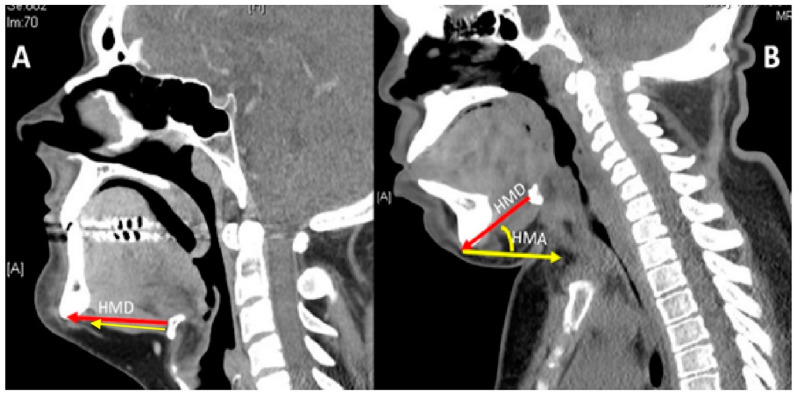
Shows the hyomental distance (HMD) and hyomental angle (HMA) in adult non-MPS marked (**A**) on the left and adult MPS patient marked (**B**) on the right, taken from a computer tomography (CT) scan of the neck in the neutral position. Red arrow represents the HMD and yellow arrow represents the horizontal.

**Figure 6 jcm-10-04924-f006:**
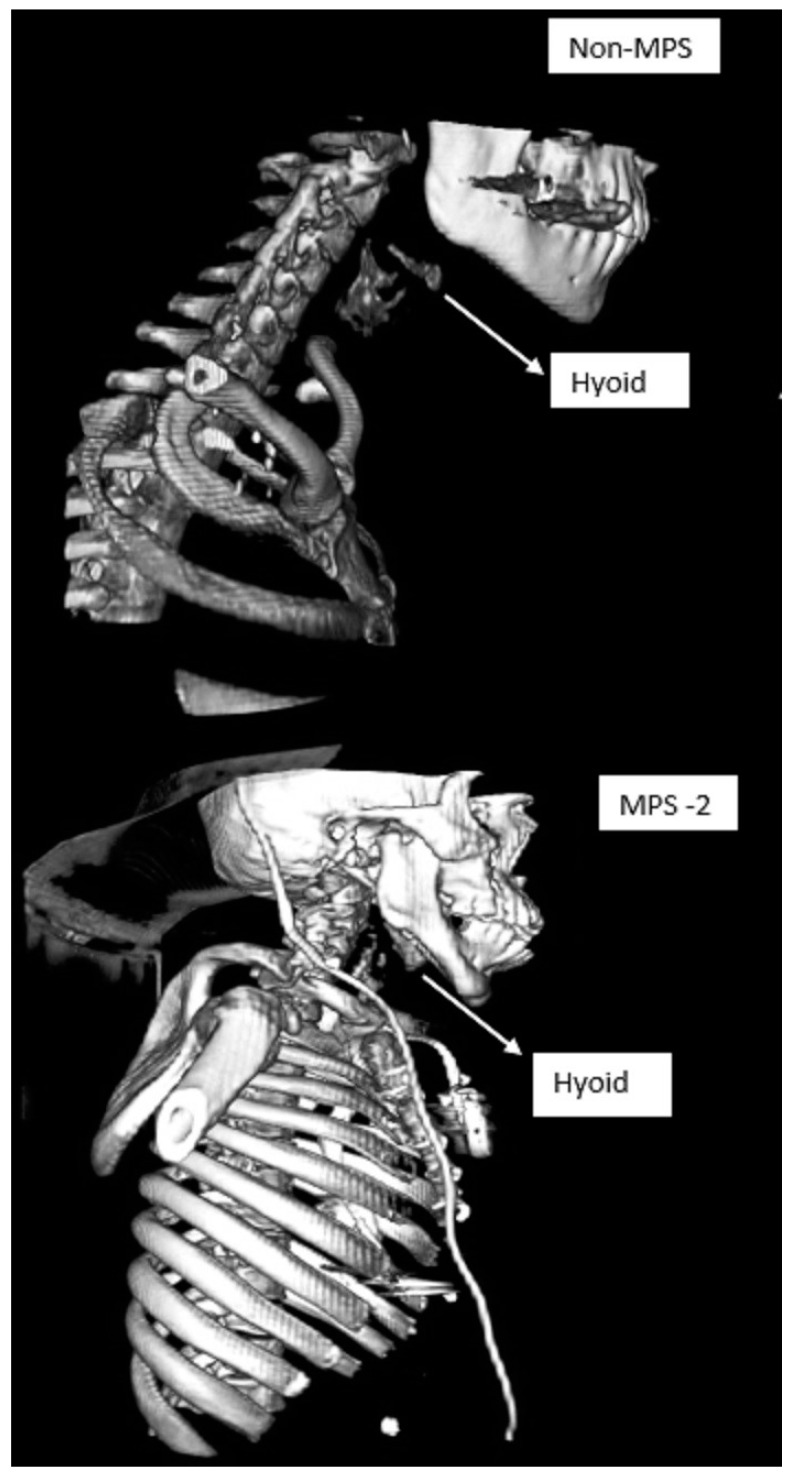
Three-dimensional reconstruction of the upper body skeleton, showing the level of hyoid bone in a non-MPS (top panel) and MPS-2 patient (lower panel) in relation to the mandible. A ventriculoperitoneal shunt and pacemaker are seen in the MPS patient. MPS- Mucopolysaccharidosis.

**Table 1 jcm-10-04924-t001:** Various types of MPS; reproduced with permission from Braunlin et al. [8], who compiled data from Neufeld et al. [9] and Valayannopoulos et al. [10]. AR: autosomal recessive; CS: chondroitin sulfate; DS: dermatan sulfate; GAG: glycosaminoglycan; H: Hurler syndrome; HS: heparan sulfate; H-S: Hurler–Scheie syndrome; KS: keratan sulfate; S: Scheie syndrome; XR: X-linked recessive. * only 1 patient reported in the literature (Natowicz et al. 1996); ** death can occur in utero with hydrops fetalis.

MPS Type (Eponym)	Incidence per 10^5^ Live Births; Inheritance Pattern	Typical Age at Diagnosis	Typical Life Expectancy If Untreated	Enzyme Deficiency	GAG
MPS I Hurler (H) MPS I Hurler-Scheie (H-S) MPS I Scheie (S)	0.11–1.67; AR	H: <1 year H-S: 3–8 years S: 10–20 years	H: death in childhood H-S: death in teens or early adulthood S: normal to slightly reduced lifespan	α-L-iduronidase	DS, HS
MPS II (Hunter)	0.1–1.07; XR	1–2 years when rapidly progressing	rapidly progressing: death < 15 years slowly progressing: death in adulthood	iduronate-2-sulfatase	DS, HS
MPS III (Sanfilippo) A-B-C-D	0.39–1.89; AR	4–6 years	death in puberty or early adulthood	heparan sulfamidase (A) N-acetyl-α-D-glucosaminidase (B) acetyl-CoA-α-glucosaminidase N-acetyltransferase (C) N-acetylglucosamine-6-sulfatase (D)	HS
MPS IV (Morquio) A-B	0.15–0.47; AR	1–3 years	death in childhood- middle age	N-acetylgalactosamine-6-sulfatase (A) β-galactosidase (B)	CS, KS (A) KS (B)
MPS VI (Maroteaux-Lamy)	0–0.38; AR	rapidly progressing: 1–9 years slowly progressing: >5 years	rapidly progressing: death in 2nd–3rd decade slowly progressing: death in 4–5th decade	N-acetylgalactosamine-4-sulfatase	DS
MPS VII (Sly)	0–0.29; AR	neonatal to adulthood	death in infancy-4th decade **	β-D-glucuronidase	CS, DS, HS
MPS IX (Natowicz) *	unknown	adolescence	unknown	hyaluronidase	CS

**Table 2 jcm-10-04924-t002:** Demographics of the study in both the MPS and non-MPS groups.

	MPS	Non-MPS
**Number of patients**	50	50
**Males**	32	25
**Females**	18	25
**Age range in years**	19–66	22–84
**Mean age in years**	31.7	59.9
**Mean Body Mass index**	26.8	26.5

MPS—Mucopolysaccharidosis.

**Table 3 jcm-10-04924-t003:** Clinical diagnosis of different patients in the MPS and non-MPS groups. MPS—Mucopolysaccharidosis.

Pathology	Number	Males	Females
**MPS group**			
MPSI	16	8	8
MPSII	13	13	0
MPSIII	1	1	0
MPSIV	14	6	8
MPSVI	6	4	2
Total	50	32	18
**Non-MPS group**			
Subglottic stenosis	10	1	9
Tracheal stenosis	8	5	3
Vasculitis	7	4	3
Malignancy not involving supraglottis, oropharynx	16	12	4
Bilateral vocal fold immobility	5	0	5
Vocal cord leukoplakia	4	3	1
**Total**	**50**	**25**	**25**

MPS—Mucopolysaccharidosis.

**Table 4 jcm-10-04924-t004:** HMD and HMA in the MPS and non-MPS groups.

**MPS**						
	**Age in Years**	**HMD in Millimeters**	**HMA in Degrees**	**Height in Centimeters**	**Weight in Kilograms**	**Body Mass Index**
*N* = 50						
Mean	31.74	46.5	27.6	135.8	50.47	26.8
Median	29.50	48.5	25.0	136.8	48	25.7
Range	19–66	25.7–66.0	−10.0 to 50	91.0–182	17.4–125.2	16.5–43.6
**Non-MPS**						
	**Age in Years**	**HMD in Millimeters**	**HMA in Degrees**	**Height in Centimeters**	**Weight in Kilograms**	**Body Mass Index**
*N* = 50						
Mean	59.9	41.9	2.420	166.1	72.6	26.5
Median	63	40.9	0.0	166.0	71.9	26.9
Range	22–99	27.0–60.3	−35.0 to 28	150.0–188	39.3–130	14.2–47.3

HMD—Hyomental distance, HMA—Hyomental angle. MPS—Mucopolysaccharidosis.

**Table 5 jcm-10-04924-t005:** Pearson correlation between HMA, HMD, height, weight, and BMI in both the MPS and non-MPS groups.

**Correlations MPS**						
		**HMD**	**HMA**	**HT**	**WT**	**BMI**
HMD	rho	1	−0.2	0.28 *	0.3 *	0.2
	*p*-value		0.15	0.05 *	0.009 *	0.13
HMA	rho	−0.2	1	−0.45 *	−0.41 *	−0.1
	*p*-value	0.146		0.0001 *	0.003 *	0.73
**Correlations Non-MPS**						
		**HMD**	**HMA**	**HT**	**WT**	**BMI**
HMD	rho	1	0.15	0.1	0.35	0.27
	*p*-value		0.3	0.49	0.014	0.06
HMA	rho	0.15	1	−0.03	−0.03	0.02
	*p*-value	0.296		0.85	0.862	0.91

* Represents significant results. HMA—Hyomental angle, HMD—Hyomental distance, HT—height, WT—weight, BMI—Body Mass Index, rho—Pearsons correlation coefficient value.

## Data Availability

All the data required to understand and the data supporting reported results this project has been provided in the paper.

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
