# Peer review of "Hyo-Mental Angle and Distance: An Important Adjunct in Airway Assessment of Adult Mucopolysaccharidosis"

_jcm, 2021, doi:10.3390/jcm10214924_

Round 1
Reviewer 1 Report
This is an interesting and very comprehensive study about airway abnormalities in adult Mucopolysaccharidosis, o-mental angle and distance which authors consider an important adjunct in airway assessment. Writing team are specialized ENTs and anaesthesiologists who put a remarkable effort into this manuscript preparation. Gathering all these data is even more appreciated as the disease is rare. Entire paper is very well written, still some very minor criticism should be addressed:
Some more keywords according to MeSH shall be added, this will improve availability through search engines and would eventually lead to improved citation odds
L52 - there are certain different ways to compare upper airways volume as well, mainly according to OSAS validation and treatment which is nowadays a hot topic. please incorporate and cite https://www.mdpi.com/2076-3417/11/9/3764
L217-218 - please provide relevant citation as this statement reads like an opinion
Author Response
We are humbled by the responses from the reviewers and we are very grateful for their opinion.
Reviewer 1:
The authors would like to thank the reviewer 1 for time, expertise and effort taken in reading our paper and offering opinion to improve.
Please find the points addressed
This is an interesting and very comprehensive study about airway abnormalities in adult Mucopolysaccharidosis, o-mental angle and distance which authors consider an important adjunct in airway assessment. Writing team are specialized ENTs and anaesthesiologists who put a remarkable effort into this manuscript preparation. Gathering all these data is even more appreciated as the disease is rare. Entire paper is very well written, still some very minor criticism should be addressed:
Some more keywords according to MeSH shall be added, this will improve availability through search engines and would eventually lead to improved citation odds
Response: Three further keywords (keyword 4 Hyoid Bone; keyword 5 Chin; keyword 6 Intubation, Intratracheal) have now been added
L52 - there are certain different ways to compare upper airways volume as well, mainly according to OSAS validation and treatment which is nowadays a hot topic. please incorporate and cite https://www.mdpi.com/2076-3417/11/9/3764
Response: This interesting article has been discussed in the section of Clinical assessment of difficult airway and has been cited accordingly.
L217-218 - please provide relevant citation as this statement reads like an opinion
Response: reference has been added for the sentence short stature and truncal obesity
Mitchell J, Berger KI, Borgo A, Braunlin EA, Burton BK, Ghotme KA, Kircher SG, Molter D, Orchard PJ, Palmer J, Pastores GM. Unique medical issues in adult patients with mucopolysaccharidoses. European journal of internal medicine. 2016 Oct 1;34:2-10.

Reviewer 2 Report
Dear Authoors,
First of all, I’ve read your manuscript gladly.
I’ve appreciated your efforts, and I think that I learned something useful for my clinical practice. Based on this paper and previous published , you’ve proven to be expert Phisiscians in MPS patients’ managing.
I’ve only few requests and suggestions in following items:
- Pag 5 line 108. In fig 5 A it could be better to insert a white arrow like in Fig 5 B, to indicating the horizontal axis in neutral position. In his case the second line is almost overlapping to HMD black arrow, so it’s easier to understand that HMA is near 0°.
- Pag 7 line 140. Please: better explain ENT acronym.
- Pag 8 line 178 (table 5) HMD doesn’t correlate with height and BMI in non MPS group, but reveals a decent correlation with height and weight in MPS (p 0.05; p 0.009). So I suggest to write that HMA shows a better correlation with height and Weight in MPS group, compared to HMD.
- Pag 10 line 248: elide The Authors because it’s a repeat.
- I think that also the neck’s maximum extent in these patients could be helpful, and must be considered. In my opinion a blocked neck doesn’t permit any correction of high HMA angle, worsening the direct or endoscopic larynx’s vision. Have you any data about this assumption?
- If you report in table 1 a very short life expectancy, especially in MPS 1-2-4; your patients’ casistic is based on less critical cases because of a medium age of 31 years. Maybe your assumption about HMA’s predictive value is even more evident in young age patients and in more severe cases of MPS.
- Finally I wonder if you could give a sort of “HMA value” prediction score: to having a good correlation between the HMA’s measure and the Cormack vision; or at least a sort of “cut off” value to alert the Anesthetist in preoperative valuation.
Best regards
Author Response
We are humbled by the responses from the reviewers and we are very grateful for their opinion.
Reviewer 2:
We would like to thank the reviewer 2 for reading the paper and offering valuable advice. We also like to thank the reviewer for finding our paper interesting and useful. We have addressed all the questions raised and please find the responses.
Dear Authoors,
First of all, I’ve read your manuscript gladly.
I’ve appreciated your efforts, and I think that I learned something useful for my clinical practice. Based on this paper and previous published , you’ve proven to be expert Phisiscians in MPS patients’ managing.
I’ve only few requests and suggestions in following items:
- Pag 5 line 108. In fig 5 A it could be better to insert a white arrow like in Fig 5 B, to indicating the horizontal axis in neutral position. In his case the second line is almost overlapping to HMD black arrow, so it’s easier to understand that HMA is near 0°.
Response: A yellow arrow is now inserted in figure 5A such as in figure 5B to represent the horizontal and a text has been added “Red arrow represents the HMD and yellow arrow represents the horizontal”.
- Pag 7 line 140. Please: better explain ENT acronym.
Response: ENT acronym has now been expanded to Ear, Nose and Throat
- Pag 8 line 178 (table 5) HMD doesn’t correlate with height and BMI in non MPS group, but reveals a decent correlation with height and weight in MPS (p 0.05; p 0.009). So I suggest to write that HMA shows a better correlation with height and Weight in MPS group, compared to HMD.
Response: The following sentence is now added “HMD shows no correlation with height, weight and BMI in non-MPS group, but reveals a statistically significant correlation with height and weight in MPS (p= 0.05; p=0.009), it must be noted that the rho value in the MPS group is only 0.3 at best. Pearsons correlation between HMA versus height, weight showed moderate negative correlation in the MPS group and no correlation in the non-MPS group. So HMA shows a better correlation with height and weight in MPS group, compared to HMD”.
- Pag 10 line 248: elide The Authors because it’s a repeat.
Response: The repetition has been deleted and sentence is now corrected to “The authors conclude that a sternomental distance of 13.5 cm or less was 66.7% sensitive, 71.1% specific, the positive and negative predictive values were 7.6% and 98.4%, for difficult laryngoscopy”.
- I think that also the neck’s maximum extent in these patients could be helpful, and must be considered. In my opinion a blocked neck doesn’t permit any correction of high HMA angle, worsening the direct or endoscopic larynx’s vision. Have you any data about this assumption?
Response: We have now included this point in the paper. We unfortunately do not have any data on measurement of the neck flexion or extension as we tested HMD and HMA in neutral position. Future studies could incorporate measurement of HMA and HMD in in neutral and extension of the neck. This is now discussed in the limitations section 4.3.1 as
“Extension of the neck will improve laryngoscopic views. Future studies could also incorporate measurement of HMA, HMD in maximum extension and comparison with HMA, HMD in neutral position to assess the degree of improvement of laryngoscopy views by neck extension”.
- If you report in table 1 a very short life expectancy, especially in MPS 1-2-4; your patients’ casistic is based on less critical cases because of a medium age of 31 years. Maybe your assumption about HMA’s predictive value is even more evident in young age patients and in more severe cases of MPS.
Response: A sentence in the discussion is now added in section 4.1
“Advances in treatment strategies has improved life expectancy, the average age of our cohort is 31.7 years. So, our study may be a representative of varied MPS phenotypes and younger adult MPS patients”.
- Finally I wonder if you could give a sort of “HMA value” prediction score: to having a good correlation between the HMA’s measure and the Cormack vision; or at least a sort of “cut off” value to alert the Anesthetist in preoperative valuation.
Response: In our personal experience in dealing with adult airways in MPS and non-MPS we feel any HMA close to zero or less than zero indicates a larynx which is not high. However, HMA should be considered as another adjunct in upper airway assessment; holistic airway assessment should include both upper and lower airways.
The following sentences have been added in the discussion
“In our personal experience of adult airway assessment, we feel HMA close to zero or less than zero indicates a larynx which is not high. HMA could be considered as another important adjunct in upper airway assessment; however, holistic airway assessment should include both upper and lower airways.
Best regards
